# Production of Coffee Cherry Spirits from *Coffea arabica* Varieties

**DOI:** 10.3390/foods11121672

**Published:** 2022-06-07

**Authors:** Patrik Blumenthal, Marc C. Steger, Andrès Quintanilla Bellucci, Valerie Segatz, Jörg Rieke-Zapp, Katharina Sommerfeld, Steffen Schwarz, Daniel Einfalt, Dirk W. Lachenmeier

**Affiliations:** 1Coffee Consulate, Hans-Thoma-Strasse 20, 68163 Mannheim, Germany; patrik.blumenthal@live.de (P.B.); marcsteger2@googlemail.com (M.C.S.); schwarz@coffee-consulate.com (S.S.); 2Yeast Genetics and Fermentation Technology, Institute of Food Science and Biotechnology, University of Hohenheim, Garbenstrasse 23, 70599 Stuttgart, Germany; daniel.einfalt@uni-hohenheim.de; 3Finca La Buena Esperanza, Pasaje Senda Florida Norte 124, San Salvador, El Salvador; coffeelbe@gmail.com; 4Chemisches und Veterinäruntersuchungsamt (CVUA) Karlsruhe, Weissenburger Strasse 3, 76187 Karlsruhe, Germany; valerie.segatz@stud.hs-coburg.de (V.S.); katharina.sommerfeld@cvuaka.bwl.de (K.S.); 5Hochschule für Angewandte Wissenschaften Coburg, Friedrich-Streib-Strasse 2, 96450 Coburg, Germany; 6Rubiacea Research and Development GmbH, Hans-Thoma-Strasse 20, 68163 Mannheim, Germany; joerg.rieke_zapp@yahoo.de

**Keywords:** coffee pulp, coffee by-products, sensory evaluation, fruit spirit, methanol, distillation

## Abstract

Coffee pulp, obtained from wet coffee processing, is the major by-product accumulating in the coffee producing countries. One of the many approaches valorising this underestimated agricultural residue is the production of distillates. This research project deals with the production of spirits from coffee pulp using three different *Coffea arabica* varieties as a substrate. Coffee pulp was fermented for 72 h with a selected yeast strain (*Saccharomyces cerevisiae* L.), acid, pectin lyase, and water. Several parameters, such as temperature, pH, sugar concentration and alcoholic strength were measured to monitor the fermentation process. Subsequently, the alcoholic mashes were double distilled with stainless steel pot stills and a sensory evaluation of the products was conducted. Furthermore, the chemical composition of fermented mashes and produced distillates were evaluated. It showed that elevated methanol concentrations (>1.3 g/L) were present in mashes and products of all three varieties. The sensory evaluation found the major aroma descriptor for the coffee pulp spirits as being *stone fruit*. The fermentation and distillation experiments revealed that coffee pulp can be successfully used as a raw material for the production of fruit spirits. However, the spirit quality and its flavour characteristics can be improved with optimised process parameters and distillation equipment.

## 1. Introduction

One of the most popular drinks in the world is coffee, which is consumed on a daily basis by millions of people [1]. Coffee drinking also represents an essential part of many cultures and traditions [2]. Furthermore, coffee beans are the most important food commodity and therefore it is the agricultural export commodity with the highest value [3]. In 2020, around 10.3 million tons (175 million 60 kg bags) of green coffee were produced worldwide [4]. The five main coffee-producing countries are Brazil, Vietnam, Colombia, Indonesia and Ethiopia [4]. El Salvador, a small country in Central America, produced 36,000 tons of green coffee in 2020 and is thus ranked 18th among the largest coffee producers [4].

During the primary coffee production, many by-products accumulate in huge amounts in the coffee-growing countries. The most common coffee by-products are the pulp, husk, parchment and leaf, besides silverskin and spent coffee grounds, which accumulate in the coffee-using countries [5,6,7]. Besides the coffee seeds, which are roasted and ground to make the popular hot beverage, the cherries consist of many layers and materials. More than 50% of the coffee cherry dry weight is composed of by-products, which accrue during coffee processing [1]. In the wet processing chain, coffee pulp is the major by-product that occurs [8]. The ripe cherries get mechanically crushed in a pulper to remove the outer skin and pulp from the seeds. When processing two tons of fresh coffee cherries, an accumulation of around one ton of coffee pulp is estimated [9]. Generally, the term coffee pulp refers to the skin (epicarp) and the pulp (mesocarp) [1]. The moisture content of fresh coffee pulp is approximately 75% [10]. The pulp of *Coffea arabica* has a pH value of 4.25 ± 0.09 and a dry matter of 22.95 ± 0.49% [11]. Braham and Bressani [8] report that dry coffee pulp contains 12.4% reducing sugars, 6.5% pectic substances and 1.3% caffeine. Coffee pulp consists of fibre, carbohydrates and proteins, but it is also rich in polyphenols, caffeine and tannins [12].

Considering the world production of green coffee, massive amounts of biomass are generated annually. The flesh of the fruit, also called pulp, is basically discarded or rarely used as organic fertilizer, but some nutrients in the fruit make the raw material interesting for further processing. The use of the coffee pulp would be essential from an environmental perspective as well. If this by-product is improperly disposed of into rivers and lakes, it can have a dramatic impact on aquatic flora and fauna [13].

For years, the price of green coffee has been at an alarmingly low level. One reason for the unpredictable price fluctuations is the international trading on the stock exchange. According to the International Coffee Organization, the average price (ICO composite indicator) for raw coffee was 2.71 $ per kg in April 2021 [14]. To cover all the costs of production, this price is barely sufficient. Due to this unstable and low price, many coffee farmers in the tropics are facing an uncertain future [15]. In addition, climate change and rising temperatures are threatening the production of premium coffee in Latin America, for instance [16]. Since green coffee prices are very low, the livelihoods of many coffee producers around the world are threatened. Therefore, it is important to rethink the utilisation and valorisation strategies of coffee by-products.

For some years now, more and more research has been conducted on new types of spirits that use unusual raw materials. If organic coffee waste from primary production can also be used for this purpose, both the valorisation and sustainability of coffee production could be increased. Due to the high concentration of carbohydrates and fermentable sugars in coffee pulp, this residue can be considered as a promising substrate for ethanol production [17]. The literature indicates great potential for spirits made from coffee by-products for human consumption, such as from coffee cherries [18], dried fruit skin [19] or fresh coffee pulp [20]. Despite its promising potential, the elevated methanol content in the spirits are problematic, for example for authorised marketing in the European Union [18]. Therefore, studies tried to find solutions to reduce the methanol content in order to obtain a safe and high-quality distillate. They showed that utilisation of different fresh fruit substrates, use of specific yeasts (i.e., strains with low capacity of producing pectin methylesterase), modern distillation or proper distillation techniques could reduce the methanol concentration in fruit spirits [21].

The main objective of this research project was to produce a fruit spirit from the pulp of coffee cherries and to evaluate its sensory quality.

## 2. Materials and Methods

### 2.1. Raw Material Collection

The practical on-site experiments for this research project were conducted from January to March 2021 in Chinameca, San Miguel, El Salvador. Ripe *Coffea arabica* L. cherries of three different varieties were collected by hand during the main coffee harvest in early 2021. The photographs in Figure 1 show the fully ripe coffee fruits on the plant branches just before harvest and the cherries in the collection tanks after picking. In these photos, the different varieties used for this scientific work can be compared with the specific appearance of the plants and fruits. While Yellow Colombia (YC) and Yellow Icatú (YI) have yellow peel colour, the exocarp is red in the Red Bourbon Tekisic (BT) variety.

The varieties YC and YI grow at an altitude of 1450 metres above sea level (MASL) and were picked on 17 February 2021. One week later, on 23 February, the variety BT was harvested; the lot is located at an elevation of 1.600 MASL. The coffee was collected and processed in a family-owned farm on the northern hillside of mountain Cacahuatique in Morazán region, El Salvador. The processing plant with the wet mill is located on 1.200 MASL (coordinates: 13°46′46.2′′ N 88°12′12.6′′ W), whereas the fermentation and distillation was carried out in the village Chinameca at an altitude of 600 MASL. The harvested coffee cherries were in an excellent optical quality, with no crushed or rotten fruit present.

Ten cherries of each variety were drawn for later physical measurements. The fruits were measured, weighed, and the pulp to fruit and the seed to fruit ratio was calculated. For exact weight measurement, a precision balance with 0.01 g scale (model CS, Reteck Electronic Co., Ltd., Shenzhen, Guangdong, China) was used. Furthermore, the fruits were cut in half with a razor blade to photograph the cross sections. A hand-held refractometer (iTavah, Guangzhou Juanjuan Electronic Technology Co., Ltd., Guangzhou, Guangdong, China ) was used to measure the sugar concentration of the pulp in Brix degrees (range: 0–32 °Bx).

After harvest, the fruits were manually screened to recover only fully ripe red and yellow coffee cherries and then washed twice with clean water. Cleaned fruits were subsequently crushed in the wet mill to separate the seeds and pulp. To avoid contamination with recycled process water, the coffee cherries were depulped in a dry manner, without water being added to the pulper machine. The pulp was recovered directly at the outlet of the coffee wet mill, and 10–17 kg was inserted into 60 L plastic fermentation barrels (Salvaplastic SA de CV, La Libertad, El Salvador).

Additionally, samples from some further coffee varieties were investigated. A detailed description of the further samples is available in Table A1.

### 2.2. Mash Preparation

As previously described, the main substrate for fermentation was fresh coffee pulp. Due to the unexpected low moisture content in the fruit flesh, water had to be added to the coffee pulp in similar quantities. Spring water from the Cacahuatique Mountain was filled in the barrels until the surface of the coffee pulp was reached. To prepare the mash, some additives (Table 1) were mixed into the coffee pulp using an electric mixing drill. The composition of the three mashes was done using similar proportions.

The pH of the mash was lowered to 3.0 with lactic and malic acid (product no. 5850, Schliessmann, Schwäbisch Hall, Germany) in order to prevent uncontrolled growth of undesirable microorganisms. BT mash was acidified with sulphuric acid. The added amount of acid was between 19 and 22 mL per kg mash. Pectin degrading enzymes were added to enhance the liquefaction of the coffee pulp. 1–2 mL of pectin lyase Ultra-Fruit (product no. 5015, Schliessmann, Schwäbisch Hall, Germany) were added to the fruit mashes. Subsequently, a selected yeast strain of *Saccharomyces cerevisiae* (AROMA Plus, product no. 5828, Schliessmann, Schwäbisch Hall, Germany) was added to initiate the controlled fermentation process. The added quantity of acid, enzyme and yeast was carried out according to the manufacturer’s instructions. The barrels were closed with a lid with rubber seal and a metal snap-closure. Air-locks filled with water were mounted on the lids to allow the escape of gases. The barrels were placed outside in the shade as previous temperature measurements showed that the average temperatures were lower outside than in the building. The next day, after 24 h, the pH was controlled and readjusted to 3.0.

### 2.3. On-site Fermentation Analysis

During the fermentation process, the ambient air temperature and mash core temperature were measured every 10 min. A thermometer with data logger (SD-947, REED Instruments, Wilmington, NC, USA) was connected with two temperature sensors. A liquid sample of the mashes was taken daily to analyse the fermentative process parameters. With a pocket-size pH tester (HI98127, Hanna Instruments, Vöhringen, Germany) the pH value of the fermented coffee pulp was measured. At the beginning of the analyses, the pH-measuring device was calibrated with two buffer solutions and then checked regularly.

A special suitcase-sized instrument (Alcoquick 4000, UNISENSOR Sensorsysteme GmbH, Karlsruhe, Germany) for wine and grape mash measurement was available on-site in El Salvador to analyse the coffee pulp mashes. The device was equipped with an infrared spectrometer and an electronic oscillation-type density meter, which facilitated the measurement of alcoholic strength, extract and density (see [22]). These fermentation parameters were determined at 24 h intervals. The mash samples had to be filtered and degassed with a lab filter before approximately 40 mL of liquid were pumped into the instrument. The mash analysis was conducted in triplicate. In addition, samples of the mash were taken on a daily basis and frozen for later high-performance liquid chromatography (HPLC) analysis in Germany.

### 2.4. Distillation Process

Distillation was performed after 72 h fermentation time, because fermentation was already finalized (see process control results below). The total weight of the three mashes after fermentation were 32.6 kg (YC), 18.8 kg (YI) and 31.1 kg (BT). The production of coffee pulp spirits was performed in a double pot still distillation process. Two different stainless steel pot stills were used. A 50 L pot still (Vevor, Shanghai Sishun Machinery Equipment Co., Ltd., Shanghai, China) was available on-site to perform raw spirit production. A 2 L pot still (Beifuss, Herborn, Germany) was used to redistill the raw spirits. The condensers were cooled with ice-chilled water that circulated in the system with a small water pump. The mash from the barrels was added to the 50 L still and heated with a gas-powered cooking plate.

The produced raw spirit was subsequently distilled with the 2 L pot still. Here, the produced distillate was fractionated into heads, hearts and tails by sensory evaluation. Every 50 mL a new glass was placed under the outlet to allow the fractionation.

In order to monitor the distillation process, the temperature, volume of outflow and alcoholic strength were recorded every five minutes. The analogue thermometer was located in the helmet of the stills. Furthermore, the flowrate of the distillates in mL/h was measured. The alcoholic strength at the outlet pipe was measured with a hand-held refractometer (0–80% vol).

### 2.5. Chemical Analysis

When the experiments were conducted in El Salvador, samples from mash and from the products were taken for the chemical analyses in Germany. For the fermentation analysis, every 24 h a 50 mL screw cap tube (Greiner Bio-One, Frickenhausen, Germany) was filled with the coffee pulp mash, properly closed and directly deep-frozen. In addition, residues that remained in the 2 L pot still after distillation were also sampled. The samples were transported to Germany by air cargo in a frozen state without being thawed.

In the laboratory of the University of Hohenheim the frozen samples were defrosted. For analysis, the liquid was pressed out of the mash and further filtered through a 0.45 µm syringe filter. 700 µL of each liquid mash sample was pipetted into HPLC analysis vials in triplicates.

To determine the fermentation process, a high-performance liquid chromatography instrument with refractive index (RI) detection was used. With the HPLC-RI instrument several compounds of the mash were analysed in triplicate, such as ethanol, methanol, glucose, fructose, sorbitol, glycerol, lactic acid and acetic acid. For the quantification of the compounds, a Rezex ROA Organic Acid H+ (8%) column (Phenomenex, Aschaffenburg, Germany) was used in combination with a refractive index detector (Shodex RI-101, Thermo Fisher, Waltham, MA, USA). Analyses were carried out by applying an operation temperature of 80 °C at the column. As eluent, sulphuric acid (0.005 N) with a flow rate of 0.6 mL/min was chosen. To identify the compounds, the retention times were compared with the measured retention times of the certified standards. With a suitable five-point calibration curve (R^2^ ≥ 0.99) of different standard concentrations, the eight substances could be quantified.

The alcoholic strengths of the distillates and their different fractions were measured using an electronic oscillation-type density meter (DMA 35, Anton Paar GmbH, Ostfildern, Germany).

For the determination of volatile compounds in the different distillates and fractions, a previously described procedure using headspace gas chromatography (GC) with a flame ionization detection (FID) was applied [18]. The following volatile compounds were analysed and quantified with the headspace GC: methanol, acetaldehyde, ethyl acetate, isoamyl alcohol (3-methyl-1-butanol), 1-propanol, 1,1-diethoxyethane, isobutanol (2-methyl-1-propanol), 2-methyl-1-butanol and 2-butanol. These substances were measured in the head, heart and tail fractions of the distillates; the concentrations are indicated in grams per hectoliter of pure anhydrous (100% vol) alcohol (g/hL pa).

In addition to the GC analyses, the heart fractions (middle fractions) of the distillates were also analysed using a nuclear magnetic resonance (NMR) spectroscopy-based spirit drinks screener [23]. The samples were measured in triplicate. The NMR method was also used to analyse the further samples from other varieties (Table A1).

### 2.6. Sensory Evaluation

The sensory evaluation was performed about ten weeks after production. The aim of the sensory evaluation was to find descriptive terms and to evaluate likeliness of the novel coffee fruit spirits. Twelve people (age 25–48) with experience in spirits tasting participated in this sensory test. All respondents have consented to participation in the study. The flavour evaluation of the spirits was performed using the method of quantitative description analysis (QDA) [24]. In addition to the QDA, an acceptance test of the spirits (total performance) was carried out as well as a multiple-samples ranking test.

In advance, the distilled spirits were diluted to 40% vol alcohol with filtered water. 30 mL of each sample was filled into small glass bottles with plastic screw caps. The bottles with the three different spirit varieties were marked with randomised three-digit numbers (YC-142, YI-423, BT-708). Each of the twelve testers was equipped with an equal amount of sample (3 × 30 mL), suitable glasses and a printed test sheet (sensory instruction sheet). The spirit samples were allowed to reach room temperature. Except for water, no food or drink was consumed 30 min prior to the test.

First, all testers tasted the first of the three samples individually and noted all perceived attributes of odour and taste into the protocol. Then, all individual terms were collected in a spreadsheet and discussed within the panel. Synonyms and hedonic terms were withdrawn from the group result to obtain a meaningful collection. Afterwards, the panelists voted for the importance of particular terms. Only if a term received the majority (>50%) in the panel was it accepted as a specific descriptor of the flavour profile. The participants then indicated the intensity of the selected terms on an ordinary intensity scale of 0–5 (not perceptible—very strong). Afterwards, the two other samples were tasted and evaluated according to the same procedure.

Following the QDA, the total performance (likeliness) should be rated on a scale of 0–5 (dislikes strongly—likes strongly) for each of the three samples. Finally, the panelists were asked to order the three spirit samples according to their personal preference of odour and taste (lowest—medium—highest). This multiple-samples ranking test assesses the differences in sensory preferences and is a simple and useful tool for that [25]. The Friedman test was chosen for the statistical analysis. Referring to ISO 8587, this test has the greatest potential to detect differences between the samples in the perception of the panelists [26].

### 2.7. Statistical Evaluation

The software Design Expert was used to statistically analyse the data from the various experiments. As a statistical analysis method, the analysis of variance (ANOVA) was applied.

## 3. Results

### 3.1. Coffee Cherry Characteristics

Figure 2 shows macro pictures of the cross sections of the coffee fruits on graph paper and thus illustrates the proportion of seeds and pulp.

Ten coffee cherries of each variety were randomly selected and physically analysed, see Table 2. The data of the physical characteristics show, that the BT variety had bigger fruits (1.53 ± 0.09 cm) than YC (1.35 ± 0.12 cm) or YI (1.38 ± 0.11 cm).

### 3.2. Fermentation Process

Monitoring the temperature data gives information about the progress of the mash fermentation and the activity of the yeast. During the fermentation of mashes YC and YI the maximum ambient air temperature was 32.5 °C, the minimum was 18.0 °C and the mean was 25.3 ± 3.3 °C. Unfortunately, the core temperature of mash YI could not be recorded, but it can be assumed that the data are similar to that of mash YC. The maximum core temperature of mash YC was 31.3 °C, the minimum was 22.5 °C and the mean was 27.0 ± 2.5 °C. During the fermentation of mash BT the maximum ambient air temperature was 33.9 °C, the minimum was 21.2 °C and the mean was 27.7 ± 3.5 °C. The core temperature of mash BT had a maximum value of 31.2 °C, a minimum value of 23.2 °C, and a mean of 28.1 ± 2.5 °C.

### 3.3. Mash Analysis

The initial pH values before the acidification of the coffee pulp mash were 4.8 (YC), 4.9 (YI) and 3.6 (BT). Even if the pH was lowered to 3.0, the value rose again the next day. Therefore, it became necessary to add acid again after approximately 24 h in order to maintain the pH value of 3.0. After that, the pH value remained stable until the end of fermentation.

At the beginning of the fermentation process, the sugar concentration in the fresh fruit mash was measured with the refractometer. This resulted in values of 8.5 °Bx (YC), 8.0 °Bx (YI) and 9.0 °Bx (BT).

The initial extract concentration of 98.0 ± 0.4 g/L in mash YC decreased to 53.0 ± 1.5 g/L within two days. After these two days, however, the alcoholic strength rose from 0% vol to the maximum value of 3.51 ± 0.08% vol. On the third and last day of fermentation, the extract concentration dropped slightly to 50.3 ± 0.4 g/L and the alcoholic strength rose to 3.29 ± 0.02% vol. The fermentation progress of the coffee pulp mashes YI and BT is fairly similar to mash YC. The data indicated that the greatest metabolic processes occur in the mashes within the first two days. After 48 h, most sugars had been degraded and much alcohol had already been formed, so that the difference to the measurement after 72 h was only marginal.

The analysis with the HPLC instrument gives a comprehensive overview about the fermentation progress of the coffee pulp mashes. As expected, when glucose and fructose concentrations decreased, the alcoholic strength in the mash increased.

Most of the monosaccharides of mashes YC and BT had already been metabolised by the second day, thus there was barely any change by the third day. In contrast to mashes YC and BT, the alcoholic strength of the YI mash still increased until the third day, although the fermentable sugars had already been metabolised on the second day. At the beginning, the glucose and fructose concentrations of mash YC were 23.09 ± 0.09 g/L and 31.64 ± 0.31 g/L. The second day, after approximately 48 h, the concentrations of glucose and fructose dropped to 2.67 ± 0.18 g/L and 1.81 ± 0.34 g/L. A similar degradation of the fermentative sugars was measured for the mashes YI and BT, while the initial glucose concentration of 30.30 ± 0.56 g/L and 23.56 ± 0.53 g/L decreased to 2.65 ± 0.32 g/L and 1.53 ± 0.03 g/L, respectively, until the second day. The fructose concentration decreased from 43.41 ± 0.76 g/L (YI) and 40.99 ± 1.03 g/L (BT) to 1.70 ± 0.43 g/L and 2.71 ± 0.11 g/L, respectively.

The three mashes from the third fermentation day show only slight differences in alcoholic strength. The two yellow-fruited varieties YC and YI have a similar ethanol content of 27.34 ± 0.19 g/L and 27.18 ± 0.44 g/L, whereas the mash of BT yields an ethanol concentration of only 25.16 ± 0.07 g/L. As expected, the glucose and fructose decrease correlated with the increasing ethanol concentration in all three mashes.

Regarding the results of HPLC analysis of the three coffee pulp mashes for the substances sorbitol, glycerol, lactic acid, acetic acid and methanol, some of these substances increased during the fermentation of the coffee pulp, and the progression over the three days is clearly detectable. The carbohydrates sorbitol and glycerol, for instance, exceed the initial concentration many times over.

In case of the YC mash, the initial concentration of sorbitol and glycerol are 0.0 g/L and 0.91 ± 0.02 g/L; after three days of fermentation, the values increase to 3.70 ± 0.01 g/L and 3.75 ± 0.03 g/L, respectively. Similar behavior is observed in the two other mash samples, YI and BT. On the third day, the concentrations 2.94 ± 0.11 g/L and 3.28 ± 0.10 g/L (YI) were measured for sorbitol and glycerol, as well as 3.84 ± 0.10 g/L and 3.25 ± 0.02 g/L (BT).

At the beginning of fermentation, no acetic acid was detected in any of the three mashes. However, by the end of fermentation after three days, there has been some formation of this substance. Consequently, 0.39 ± 0.03 g/L acetic acid is measured in mash YC, 0.38 ± 0.12 g/L in YI and 0.30 ± 0.02 g/L in BT. On the other hand, higher concentrations of lactic acid were measured in the mashes. In the YC sample, 2.78 ± 0.27 g/L were present at the beginning and 2.75 ± 0.01 g/L at the end of fermentation. Higher concentrations of lactic acid were detected in the YI mash with 4.30 ± 0.08 g/L (day 0) and 4.18 ± 0.09 g/L (day 3). The BT mash has noticeably lower contents, where 0.91 ± 0.16 g/L and 0.84 ± 0.01 g/L lactic acid were determined at the beginning and end of fermentation.

The measurement of methanol showed a similar concentration in all three samples of freshly produced mashes (day 0), with 0.50 ± 0.09 g/L, 0.43 ± 0.07 g/L and 0.29 ± 0.05 g/L (YC, YI, BT). The methanol content increased during the fermentation process until the highest value was measured on the third day. Before distillation, there were 1.61 ± 0.02 g/L, 1.60 ± 0.19 g/L and 1.33 ± 0.12 g/L methanol in the three coffee pulp mashes (YC, YI, BT).

### 3.4. Distillation Process and Analysis

As presented in Table 3, the first distillation step of the coffee pulp mashes took 142 min (YC), 110 min (YI) and 184 min (BT). The generated yields and alcoholic strengths of raw spirit from the different mashes were 3600 mL at 20% vol for YC, 2700 mL at 18% vol for YI and 3000 mL at 23% vol for BT, respectively.

The diagrams in Figure 3 indicate the distillation process of temperature, alcoholic strength and product yield of the second distillation run. As the volumes of raw spirit were too large, the small 2 L pot still had to be operated twice for the distillates of each variety. Therefore, in the following diagrams only the first of both distillation runs for the fine spirits are presented. When comparing the graphs of the three experiments a rather similar progression can be seen. The temperature in the helmet of the still was 72 °C (YC), 74 °C (YI) and 75 °C (BT), respectively, after the first distillate drops ran out of the pot still. By the end of distillation, the temperature had slowly risen to 92, 95 and 92 °C. The three distillates, YC, YI and BT initially had an alcoholic strength of 62, 60 and 61% vol; when the distillation process was finished after 55–65 min, the concentrations were 41, 21 and 42% vol, respectively. After 55 min of distillation, the volumes of the produced distillates were 420 mL (YC), 450 mL (YI) and 435 mL (BT). The yield of head, heart and tail fractions can be seen in Table 4. The product flow rates of the fine spirits in the 2 L still were 461.5 mL/h (YC), 443.1 mL/h (YI) and 505.3 mL/h (BT).

Table 4 presents the collected data of volume, weight and alcoholic strength of the head, heart and tail fractions. The heart fractions of the coffee pulp distillates contained an alcoholic strength of 64.8% vol (YC), 61.8% vol (YI) and 66.1% vol (BT). The yield of heart fractions were 600 mL (YC), 500 mL (YI) and 700 mL (BT), the volume of head fractions were 150 mL (YC), 70 mL (YI) and 92 mL (BT).

Figure 4 shows the concentration and composition of the volatile compounds detected in the coffee pulp distillates. The concentration of ethyl acetate, isoamyl alcohol (3-methyl-1-butanol), 1-propanol, acetaldehyde, 1,1-diethoxyethane, isobutanol (2-methyl-1-propanol), 2-methyl-1-butanol and 2-butanol decreased in the process of distillation. This drop in concentration is an indication of the effective separation of the heads from the middle fraction (hearts).

In the shift from heads to hearts, the acetaldehyde content decreased from 119 to 28 g/hL pa, 113 to 32 g/hL pa and 262 to 58 g/hL pa (YC, YI, BT). A similar degradation was measured for ethyl acetate for the three varieties YC, YI and BT when the content decreased from 537 to 72, 226 to 40 and 366 to 56 g/hL pa. The higher alcohols such as isoamyl alcohol (235, 161, 186 g/hL pa), isobutanol (50, 39, 39 g/hL pa) and 2-methyl-1-butanol (28, 20, 30 g/hL pa) were found in higher quantities in the heart fractions (YC, YI, BT). Further volatile compounds were identified in the heart fractions of YC, YI and BT distillates, 1-propanol with 134, 117 and 119 g/hL pa, 2-butanol with 0.3, 0.2 and 1.5 g/hL pa and 1,1-diethoxyethane with 9, 13 and 26 g/hL pa.

Only the methanol concentration increased from head to heart to tail fractions. Methanol, a substance that is harmful to human health in certain quantities, was found in elevated concentrations in all distillates. The identified level of methanol in the heart fractions were 1565, 1645 and 1516 g/hL pa (YC, YI, BT).

Most of the analysed substances were also found in the tail fractions and the raw spirit residue in the still, except 2-butanol and 1,1-diethoxyethane.

Table 5 presents the results of the NMR spectroscopy analysis as mean value of the triplicate determination with the corresponding standard deviation. The volatile compounds methanol, ethyl acetate, acetaldehyde, isobutanol, 1-propanol, isoamyl alcohol, phenethyl alcohol and ethyl lactate were determined in the heart fractions of the distillates from the three varieties. The NMR results of samples from further varieties are presented in Table A2.

The comparison of the different process parameters indicates which method has a significant effect on the methanol content in the coffee fruit distillate. The ANOVA analysis shows that there is a highly significant influence of the coffee variety (*p* < 0.0001) and the raw material (*p* < 0.0001) on methanol concentration. Otherwise, no significant impact of fermentation time, acid addition, enzyme addition or selected yeast strain could be detected. Furthermore, the ANOVA evaluation also showed that the coffee variety has a significant (*p* = 0.0094) influence on sensory preference.

### 3.5. Sensory Evaluation

The major flavour descriptors of the fruit spirit from the variety YC with highest intensity scores for odour were *stone fruit*, *earthy* and *woody*. However, for the taste the highest rated terms are *vegetative-earthy*, *green notes* and *sweet*.

For the YI spirit, the three main flavours were described as *stone fruit*, *plum* and *compote/jam*. Nevertheless, it is important to mention that some negative terms are also listed, such as *solvent* and *pungent*. The major flavour characteristics of taste were *sweet*, *herbs* and *dried fruits*.

The most relevant term for odour of the BT pulp spirit is *stone fruit*, followed by *cherry* and *solvent*. However, no off-flavours could be detected in the taste of the BT sample. The flavour terms with the highest intensity were *fruity*, *stone fruit* and *cherry*.

In summary, all three novel fruit spirits convinced the panelists with their odour of stone fruits and sweetish taste. Although the floral aromas did not reach the highest intensities, it is nevertheless interesting to mention them. For instance, the sample YC was described as *rose* and YI as *elderflower*. Additionally, two of the three samples also had distinct off-flavours. The odour of YI was described with the terms *solvent* and *pungent/alcoholic*, and the BT distillate was described as *adhesive* and *solvent*.

In the acceptance test with the evaluation of total performance, the three coffee pulp spirits were rated on a scale from 0–5 (dislikes strongly—likes strongly). For the evaluation, the mean value and the standard deviation of the 12 individual results were calculated. The higher the value, the more popular the product is according to the total performance. The highest rated spirit with 3.13 ± 0.77 points was produced from the *C. arabica* variety YC. This was followed by the variety BT with 2.96 ± 1.08 points and last of all YI with 2.67 ± 1.21 points.

For the last evaluation, the multiple-samples ranking test, the sums of all data were formed. Because the scale of popularity went from low, medium to high, these ratings were converted to numbers (1, 2, 3) for the statistical calculation. Through the comparison of the sums, the sample with the highest score has the highest popularity. The results for odour indicate the highest popularity for the spirit from the YC variety (27 points), followed by BT (24 points) and YI (21 points). However, the results for taste show the highest popularity for BT (28 points), then YC (25 points) and finally again YI (19 points). The comparison of rank sums clearly shows that the spirit from the variety YI is the least popular, both in odour and in taste. The multiple-samples ranking test of the three fruit spirits indicated some preferences. In order to detect possible differences between the samples in terms of preference, the Friedman test was applied. However, based on the perception of the participants, the statistical analysis has proven that there is no significantly preferred spirit sample.

## 4. Discussion

### 4.1. Raw Material

One of the first difficulties encountered in the project was the relatively dry coffee pulp. The recently harvested coffee fruits obviously contained a certain amount of moisture, but there was not as much juice in the pulp as there would normally be. The percentage is similar to the results of other scientists, who quantified approximately 75% moisture content of fresh pulp [10]. In the pulp of red coffee cherries of the variety Costa Rica (*C. arabica*), a total moisture content of 85% was measured [27]. This shows that the moisture content of the pulp measured in El Salvador was up to 11.2% lower compared to data from the literature, although these were different varieties.

According to the experienced coffee farmer and plantation manager from El Salvador, altered climate conditions might have influenced the coffee pulp composition. While in normal years there are about 2500 mm of rainfall, at the end of 2020 there were up to 5000 mm on the plantation. It is assumed that bad weather events have led to a lower moisture content due to prolonged maturing times. An analysis of the moisture content during the ripening period revealed that the relative water content in the coffee cherries decreased constantly during fruit development [28].

Esquivel et al. [29] have evaluated varieties with different peel colors with regard to the secondary plant metabolites. The color of red-peeled coffee cherries is due to anthocyanins and carotenoids, while the color of yellow-peeled cherries is developed only from carotenoids [29]. There is also evidence that the beans from yellow coffee fruits contain more glucose, fructose and sucrose than the ones from red fruits [30]. This finding correlates with the measured sugar concentrations of the fresh coffee pulp. Both yellow-fruited varieties Colombia and Icatú have shown higher Brix values (21.5 ± 0.5 °Bx and 24.0 ± 1.0 °Bx) than the red coffee pulp of BT (20.7 ± 1.5 °Bx). In addition, it was found that the coffee pulp of red cherries is probably more acidic than from yellow ones. Before the addition of acid, the natural pH of the fresh mashes YC and YI was 4.8 and 4.9, whereas the pH of BT was only 3.6.

As the experiments in El Salvador involved depulping the coffee cherries without water, the sugar content in the pulp may be higher than described in other studies. The recirculating processing water successively enriches nutrients by diffusion from the pulp until high levels can be found in the final wastewater [17].

### 4.2. Fermentation Process

As recommended by Einfalt et al. [18], the mashing and fermentation processes took place on-site in a coffee producing country directly after the harvest. As the fruits were processed within a few hours of being harvested, the risk of quality defects such as microbial infection was presumably reduced. Since the mashes were prepared immediately after the pulping of coffee cherries, the pulp as substrate was very fresh. Nevertheless, some additionally produced mashes from the coffee pulp went moldy, especially if the coffee cherries had not been washed beforehand. In some additional experiments, the coffee pulp mashes spoiled very quickly, sometimes after only 48 h of fermentation. The surface was covered with a white mold; it is probable that the undesired microorganism was some kind of kahm yeast. It seems that the coffee pulp is an optimal breeding ground for microorganisms such as yeasts. The conditions with available nutrients (sugars), moisture and warmth appear to be beneficial for the growth of microorganisms. Microbiological tests on agar plates showed that even the pulp of the clean and washed cherries collected from the wet mill were contaminated. If the by-products of coffee production are used in the future, the processing plants such as the pulper should be technically optimised and regularly cleaned to ensure high-quality standards.

Pereira Bressani et al. [31] have investigated the influence of fermentation during natural and pulped natural processes on coffee quality. Comparing four different yeast strains (*Meyerozyma caribbica*, *Saccharomyces cerevisiae*, *Candida parapsilosis* and *Torulaspora delbrueckii*), *Saccharomyces cerevisiae* was the most suitable for *Coffea arabica* variety Mundo Novo, resulting in quality improvements and the highest sensorial scores. The fermentation of entire coffee cherries (*Coffea canephora*) inoculated with the selected yeast strain *M. caribbica* during the natural process showed the potential to increase the coffee quality [32]. Hence, it can be hypothesised that a fermentation of coffee cherries with *Saccharomyces cerevisiae* in an aqueous medium would be an alternative approach to pursue. Within the context of this research, it was not possible due to the time restrictions of necessary storage of green coffee before roasting to evaluate the coffee beans obtained during the fermentations. Nevertheless, this approach could bring the farmer double profit, on the one hand by enhancing the quality of the coffee, and on the other by obtaining a distillate obtained from the alcoholic fermentation liquid.

Due to the different coffee harvest periods, the three mashes could not be fermented at the same time. The varieties YC and YI were picked one week in advance of the red-fruited variety BT. Even though there was only one week between the two experiments, the ambient air temperature had increased. Therefore, the max, min and average temperatures were approximately 2 °C higher. However, this temperature increase raised the BT mash by only an average of 1 °C compared to the mash YC.

The manufacturer of the selected yeast strain AROMA Plus suggested a minimum temperature of 15 °C, whereas the recommended optimum temperature for the mash fermentation is in the range of 18–20 °C. The manufacturer further claims in the specifications that at higher temperatures too many volatile aromatic compounds are released by the carbon dioxide [33]. In fermentation experiments of coffee husks with *S. cerevisiae* at 25, 30 and 35 °C, the ethanol yield was best at 30 °C [34]. Although the performance was best at 30 °C, since only bioethanol and no fruit spirit was produced, the disappearance of the aromatic compounds can be neglected. A sensorial evaluation of marula fruit distillates revealed that the testers preferred the samples fermented at 15 °C over those fermented at a temperature of 30 °C [35].

### 4.3. Mash Analysis

During the analysis of the three mashes, the fermentation parameters were determined with different devices, such as the Alcoquick 4000 instrument in El Salvador and the HPLC-RI in the laboratory in Germany. With the Alcoquick device, the fermentation process could be monitored quickly and easily on-site to determine the extract and alcohol content. The instrument was an excellent help in El Salvador, because it enabled the detection of the completion of the fermentation process so that the fermented mashes could be distilled immediately. With the HPLC-RI, it was possible to quantify specific substances afterwards, such as the reduction of sugars (glucose, fructose) and the formation of alcohols (methanol, ethanol). In addition, it could be detected that sugars were not only metabolised, but that some carbohydrates were also created (sorbitol, glycerol). This could be a reason why the extract content in the Alcoquick analysis remained at 50.3 g/L (YC), 43.5 g/L (YI) and 42.6 g/L (BT) when the fermentation was completed. Nevertheless, Lachenmeier et al. measured approximately 25 g/L total dry extract in fully fermented wine with the Alcoquick 4000 [22]. Overall, the analysis results of the two measuring instruments (the Alcoquick 4000 and the HPLC-RI) clearly showed that fermentation was generally completed by the second day. The HPLC-RI results indicated that after only two days, the majority of the fructose and glucose was degraded equally in the three mashes. In contrast to mashes YC and YI, the detected lactic acid content was much lower in BT.

The addition of water was crucial for the alcoholic fermentation of the coffee pulp. However, this led to a lower sugar concentration and therefore lower alcoholic strength in the mashes. The refractometer was used to measure the sugar content in the pulps, which were 21.5 ± 0.5 °Bx (YC), 24.0 ± 1.0 °Bx (YI) and 20.7 ± 1.5 °Bx (BT). In contrast, at 8.5 °Bx (YC), 8.0 °Bx (YI) and 9.0 °Bx (BT), the fresh and still unfermented mash had a sugar content of about half. This sugar decrease is expected, because water was added to almost the same proportion of coffee pulp. Unfortunately, the addition of water also leads to an increase of energy input and distillation time. On the other hand, in previous trials, the performance of the yeast was reduced when no water was added. In Einfalt et al. [18], a sugar content of 12.2 °Bx was measured in the fresh coffee fruit mash where no external water was used.

The completely fermented mashes contained 3.29% vol (YC), 3.20% vol (YI) and 2.91% vol (BT) of ethanol. The lowest alcoholic strength was detected in the mash of BT, and this coffee pulp also showed the lowest sugar concentration. The analysis with the Alcoquick 4000 instrument indicated a slight alcohol decrease from the second to the third day in the yellow-fruited coffee pulp mashes. For instance, the alcoholic strength in mash YC decreased from 3.51 to 3.29% vol and mash YI from 3.55 to 3.20% vol. In contrast, the data from HPLC-RI measurement do not confirm the alcohol loss. In mash YI, for example, the alcoholic strength increased from 21.53 to 27.18 g/L and the methanol concentration from 1.28 to 1.60 g/L from the second to the third day of fermentation. Einfalt et al. [18] achieved an ethanol concentration of 31 g/L in the coffee cherry mash, whereas in our experiments with coffee pulp approximately 25–27 g/L were obtained.

The polysaccharide pectin is a main component of the cell walls of plants. The enzyme pectin methylesterase transforms pectin into pectic acid and methanol [36]. As fruits contain high amounts of pectin, spirits made from them (such as plums, apples or coffee cherries) also have a higher methanol content compared to distillates made from grains or sugar cane [21]. Both mashes, YC and YI, showed an almost similar methanol concentration, whereas the value of BT was lower. An equal proportion was represented in the distillates, where the sample BT achieved the lowest methanol concentration as well. A comparison of the chemical composition of different colored passion fruit varieties showed that the pectin content in yellow fruits is higher than in purple or orange fruits [37]. If this observation can be transferred to coffee cherries, it could possibly explain the slightly higher methanol content in the mashes of the yellow varieties. In this research project, the pectin content of coffee pulp was not measured. In the pulp of the *Coffea arabica* variety Catuaí 99 Vermelho, the total pectin content was 11.37 g/100 g dry matter [17]. To reduce the methanol concentration in the coffee pulp spirits, perhaps a suitable selection of raw materials could improve the result. A study proves that there are different pectin contents depending on the *Coffea arabica* variety. In the variety Bourbon, for instance, a higher pectin content was measured than in the Catimor or Caturra varieties [38]. Furthermore, there are differences in the different layers of the fruit. The coffee pulp contains approximately 1.9 times more pectin than the mucilage [38].

### 4.4. Distillation Process

The fermentation of the coffee pulp was performed with special materials that are also used in professional distillation technology, such as a selected yeast strain, acid combination, pectin lyase, air-locks or fermentation barrels. In contrast, the stainless-steel distillation equipment was neither particularly professional nor of high-quality, but it was sufficient for this first pilot experiment. To improve the flavour quality of distillates, an automated tail separation by means of in-line conductivity measurement could be a possibility [39,40]. A higher quality copper still should also be considered to create cleaner and high-grade coffee pulp spirits. Cho et al. [41] have analysed the fruit spirit characteristics by using a stainless-steel pot still and a copper multistage still. When comparing the two distillation devices, the yield of the multistage still was higher. In addition, the organoleptic quality of the fruit spirits increased using the multistage copper still due to the removal of impurities and the enhanced fruit flavours [41].

As the alcoholic strength in the mashes was quite low, the alcohol yield was consequently also not particularly high. It was possible to obtain 600 mL spirit (64.8% vol) from 32.6 kg of the mash YC, 500 mL (61.8% vol) from 18.8 kg of mash YI and 700 mL (66.1% vol) from 31.1 kg of mash BT. If the dilution to an alcoholic strength of 40% vol is taken into account, the calculated yield of fruit spirits are 942 mL (YC), 750 mL (YI) and 1117 mL (BT). Correlation between the mash weight and the alcohol recovery shows that the yield is lowest for sample YC. In addition, a higher quantity of head fraction was also separated in sample YC (150 mL) than in the other two distillation experiments YI (70 mL) and BT (92 mL). Based on these measurements, it could be assumed that the distillation process of YC was preferable, since more volatile concentrations with unpleasant aroma qualities of the heads fraction were separated from the hearts fraction. With regard to the sensory evaluation, no off-flavours were perceived in distillate YC. In contrast, the odour of the YI and BT distillates was described as *solvent* and *adhesive*, which can be caused by incorrect separation of the head fractions.

The distillation was probably not an energy-economic process, because much gas was needed for heating and then a lot of ice was needed for cooling. For only a few hundred mL yield, much energy and resources were used. The water addition therefore should only be considered as an emergency solution in special years. As mentioned earlier, the sugar and later the alcoholic strength are reduced by the addition of water.

### 4.5. Distillate Analysis

Based on the GC-FID analysis, the higher alcohols in the three distillates could be quantified. Higher alcohols contribute significantly to the aroma profile of fruit spirits. When these substances are present in small quantities in the spirits, they provide an especially pleasant taste and create an essential character [42]. A decrease of certain volatile compounds has been observed from head to heart fractions, such as acetaldehyde and ethyl acetate. In higher concentrations, both substances cause pungent characteristics in spirits [43]. Since a low concentration of acetaldehyde and ethyl acetate has a positive influence on the spirit quality, the heads should be properly separated from the hearts. Einfalt et al. [18] have detected an acetaldehyde concentration of 10 g/hL pa in the coffee cherry spirit, while 28–58 g/hL pa were measured in our coffee pulp spirits. However, in our samples the ethyl acetate concentration was with approximately 40–72 g/hL pa lower compared to 200 g/hL pa in the coffee cherry spirit [18]. In apple brandies, for instance, an ethyl acetate content of only 10.5 to 19.9 g/hL pa was analysed [44]. 1-Propanol was found in concentrations of approximately 117–134 g/hL pa in the coffee pulp spirits, whereas in cornelian cherry spirits an average 22 g/hL pa were detected [45]. The volatile compound 1-propanol has a sweetish and pleasant odour, but in excessive concentrations a solvent aroma is perceived [45].

It was not possible to determine caffeine levels with the existing equipment and methods used. Neither from the mashes nor from the distillates is data on the caffeine content available. However, this would be interesting to know, since no scientific knowledge has been gained to date. Although caffeine is easily soluble in water, it is not a volatile substance. Therefore, it is questionable whether caffeine is present in the spirit at all after distillation. In the GC-FID method, the retention time of caffeine was the same as that of ethanol, so no peak could be evaluated and ethanol content could be slightly overestimated. However in NMR, no resonances of caffeine were detectable, but the detection limit of NMR is comparably high (about 1 mg/L), so that trace caffeine levels would not be detectable. 

In the chemical analysis of the distillates, an increased methanol content was measured compared to other fruit spirits. The methanol concentrations of the heart fractions measured by GC-FID were 1565 g/hL pa for the YC variety, 1645 g/hL pa for YI and 1516 g/hL pa for BT. Even though the mean values of NMR analysis are a bit lower, it was noticed that the YI distillate has the highest and BT the lowest methanol content (YC 1412.7 ± 158.8, YI 1456.7 ± 159.8, BT 1403.7 ± 144.7 g/hL pa). In general, the three spirits have similar levels of methanol, whereas the distillate from the red-fruited variety Bourbon Tekisic has the least. These values also correspond to the mash analyses, in which the BT sample contained the lowest concentration of methanol.

The results of these experiments show that the methanol content could almost be reduced by half compared to a previously produced coffee cherry spirit with a methanol content of 2600 ± 400 g/hL pa [18]. The use of a very fresh substrate, the short fermentation period and the direct distillation after the end of fermentation may have contributed to the lower methanol content [21]. However, it must also be noted that the raw material cannot be exactly compared with each other. In this experiment, only the coffee pulp, one part of the coffee fruit, was used. On the other hand, Einfalt et al. [18] fermented mashed coffee cherries, and thus the mash included both coffee pulp and beans. As the coffee beans were still coated with the mucilage, also known as the pectin layer, the methanol content in the coffee cherry spirit may therefore be higher than in the coffee pulp spirit.

Even though the methanol content could be significantly reduced, the legal limit was still exceeded. According to the European food regulations, the maximum methanol level in fruit spirits is 1000 g/hL pa [46]. However, the spirit drinks regulation has a few exceptions for higher methanol contents of some fruit varieties. For apples (*Malus domestica* Borkh.), apricots (*Prunus armeniaca* L.) or plums (*Prunus domestica* L.), for instance, a methanol content of 1200 g/hL pa shall not be exceeded. The maximum possible content of 1350 g/hL pa applies to some fruits and berries such as quince (*Cydonia oblonga* Mill.), blackcurrant (*Ribes nigrum* L.) or elderberry (*Sambucus nigra* L.), while it is 1500 g/hL pa for fruit marc spirit [46]. Unfortunately, the methanol concentration in the coffee pulp spirits exceeds the legal tolerance by more than 50% according to the general maximum level of 1000 g/hL pa for fruit spirits, while iterating around the maximum limit for fruit marc spirit.

Despite the thoroughly positive results from the sensory examination, marketing of the novel spirits would unfortunately not be permitted in the European Union. On the one hand, the novel food approval in the European Union currently only covers dried cherry pulp and its infusion as a traditional food from a third country (Commission Implementing Regulation (EU) 2022/47). On the other hand, due to the exceeded methanol limit, it is not permitted to place the product in the market according to the EU’s spirit drinks regulation until the coffee cherry would be added to the list of fruits with higher allowed methanol contents.

The two authors of a patented method succeeded in producing a coffee fruit spirit with a methanol content below the maximum value for fruit spirits due to the artificial addition of sugar to the mash [47]. Although they described methanol concentrations of 684 and 573 g/hL pa, they were below the maximum value of 1000 g/hL pa as outlined by the European food regulations [46]. Other authors have also described the addition of a carbohydrate source (sugar, millet or rice) for ethanol production from coffee by-products; however, the methanol content was not analysed in the studies [13,19]. The addition of sugars, such as sucrose, to the mash could reduce the methanol content in the distillate to below the limit. Nevertheless, due to the addition of foreign sugar, this product would still not be allowed to be commercialised within the EU as a fruit spirit, which must be exclusively produced from fruits.

An option might be a mixture of several fruits, for example as in fruit spirits from apples and pears. There are also promising publications on tropical fruit spirits, such as a mixed spirit from passion fruit and mango [48] or banana [49]. By the addition of a fruit with low methanol formation capability (i.e., low in pectins), the resulting fruit spirits from coffee cherries and other fruits may be below the legal limit. Future research should look for a pairing with fitting sensory profiles, considering the availability of fruits on coffee plantations and similar ripeness times as coffee cherries. During the research project from January to March 2021, fruits such as citrus fruits, bananas and papayas could be harvested in the San Miguel region of El Salvador. There are also many mango trees in the area, but these fruits were still green and unripe at that time. As papaya juice was measured to have 11.11 °Bx and a sugar concentration of 41.6 g of fructose and 46.1 g of glucose per liter of juice, it can be a good substance for alcoholic fermentation [50]. A wine made from the papaya fruit (*Carica papaya*) had an alcoholic strength of 11.3% vol [51]. Another team of scientists fermented papaya juice at 29.4 °C with *S. cerevisiae* yeast and obtained a good wine with an alcoholic strength of 10.2% vol [52]. Due to the high alcoholic strength of papaya wine, this would certainly be interesting for distillation and would achieve high yields. To date, there are still no scientific articles on papaya distillates or data about the possible methanol content. Fruit spirits produced from mango and banana pulp contained a methanol concentration of 79.4 and 46.9 g/hL pa [49]. In conclusion, a mixture of coffee pulp and papaya for spirit production would be an interesting approach and the flavours of these fruits would certainly make a good combination.

Menezes et al. have studied the chemical composition of coffee pulp and its extracts. The analysis shows that the extract pressed out of the pulp contains only about 1/4 of the total pectin content as the pulp itself [53]. These results provide a promising pathway for further experimentation, and perhaps the use of the extract is a potential solution for methanol reduction. When only the pressed juice of the pulp is used for fermentation, the methanol content may be reduced compared to the coffee pulp mash produced in this experiment in El Salvador. On the other hand, it can be assumed that a part of the glucose and fructose content will remain in the press residues and is therefore not available for fermentation. Furthermore, it is probable that this has a major influence on the flavour of the spirits. Garcia et al. described the pectin extraction from coffee pulp juice and pressed pulp. From 44.84 kg fresh coffee pulp (*C. arabica* var. Bourbon) the researchers yielded 17.94 kg juice and 26.90 kg pressed pulp. After several extraction and washing steps with ethanol and acetone, 49.28 g and 109.1 g pectin were recovered from the juice and pressed pulp [38]. These data show that in the pressed pulp, the exocarp of coffee cherries contains more pectin than the fruit juice. One approach to methanol reduction in future experiments could be to ferment only the sugar-containing juice of the coffee cherry instead of the pectin-rich coffee pulp. Promising results came from the study by Zhang et al. in which the scientists fermented and distilled different plum mashes to investigate the methanol concentrations. The methanol concentration of plum spirits can be significantly reduced when using plum juice instead of plum mash as raw material with an almost equal alcohol yield [54]. The same research team conducted a similar experiment with apple spirits. The methanol content in spirits made from apple juice is lower than the distillates from apple mash [55]. However, a change in the flavour composition and sensory perception can be expected when only the fruit juice is used for the fermentation substrate and not the entire coffee pulp. A comparison of apple brandy production methods showed that a spirit distilled from apple wine reached a greater sensory score than from apple mash [56].

In fact, the addition of the enzyme pectin lyase may have been unnecessary in retrospect. Because the addition of water was mandatory due to the fermentation performance, the mash was already liquid. Therefore, the enzyme may no longer have contributed to the liquefaction, but possibly slightly to methanol formation. Nevertheless, the manufacturer of the enzyme preparation (*C. Schliessmann*) claims that distillates whose mashes have been treated with pectin lyase have significantly lower methanol content than with conventional pectinolytic enzymes.

A higher methanol concentration was detected in the tail fractions of the three samples than in the heart fractions. This behavior is similar to the distillate analysis with GC-FID by Einfalt et al. [18] and literature data on other fruit spirits [21]. Methanol has a lower boiling point than ethanol and should therefore evaporate faster. However, the actual behavior is completely different in hydroalcoholic solutions, because methanol is extremely soluble in water, so its solubility rather than its boiling point is the major influence. Thus, the distillation behavior of methanol is parallel to the one of ethanol and the compounds cannot be separated from each other during simple pot still distillation. At the end, methanol is enriched in the tailing rather than in the heads [21]. The highest methanol concentration was detected in the residue in the pot after distillation. This means that part of the methanol did not volatilize during distillation even though it has a lower boiling point than ethanol and water. This can be explained by the previously described characteristics and behavior of methanol. Thus, methanol accumulates in the pot until the very end of the distillation.

When comparing the production process of the coffee pulp spirits with the recommendations of the literature on methanol reduction, it is noticeable that many parameters have already been adjusted correctly for this research project. First, high fruit quality and freshness were emphasized because it has been proven that this quantitatively affects the formation of methanol [57]. In addition, a professional distiller’s yeast of the strain *Saccharomyces cerevisiae* was used for the fermentation of the three mashes. A comparison of eight different yeast strains indicated that *S. cerevisiae* developed the second lowest methanol concentration in the coffee pulp mash [17]. An effective approach to methanol reduction that was not carried out in these experiments is a pasteurization step. Heat treatment of the fruit mash before fermentation inactivates pectic enzymes, which can significantly reduce the methanol content in distillates [58].

According to the present state of knowledge, the authors are not aware of any scientific article that describes the production of a coffee fruit spirit that would comply with the regulations of the European food law. Either the methanol limit is exceeded in the distillates, or this value is only complied with by the non-authorised addition of sugars to the fruit mash.

### 4.6. Sensory Evaluation

In the sensory evaluation of the three coffee pulp distillates, 12 experienced testers took part. Even though the test had to be performed by video conference due to the global pandemic, the online tasting of the three spirit samples worked really well and was successful. In carrying out the quantitative description analysis, many interesting but also surprising aromas were detected.

Remarkably, there was a great similarity in the odour of the samples. The aroma descriptor with the highest rated intensity was *stone fruit* for all spirits. In addition, the descriptor *plum* was determined for odour in the three distillates made from YC, YI and BT. Further descriptive terms for the odour were *earthy*, *woody* or *floral* for YC, *elderflower*, *pungent/alcoholic* and *solvent* for YI, and *cherry*, *solvent*, and *adhesive* for BT. The odour of the yellow coffee cherry spirits was associated with more fruity terms, whereas the taste was more earthy and green. This indicates a high aroma complexity of the two spirits. The descriptors for the taste of the YC spirit were *vegetative/earthy*, *green notes*, *woody* and *rose*, whereas YI was described with the terms *herbs*, *dried fruits* and *earthy*. These results are partly consistent with the sensory evaluation of a coffee cherry spirit from Einfalt et al. [18]. The authors described the odour of the spirit as *vegetal*, *nutty* and *earthy* and the taste as *vegetal*, *alcoholic*, and *nutty.* Even though no nutty aromas were detected by the panelists in the two samples of YC and YI, the terms *earthy* and *vegetative* are matching. The taste of the spirit made of BT pulp in contrast was described as *fruity*, *stone fruit* and *cherry*. Since these terms were also used to describe the odour, they are probably the most important descriptors to characterise the spirit. It is astonishing that the term *cherry* was already used to describe a spirit made from the dried coffee pulp in literature that is barely 125 years old [59].

The two spirits from yellow coffee varieties convince with a diverse flavour profile. They are fruity in smell, but rather earthy and vegetative in taste. The red variety, on the other hand, has an intense fruity aroma in both attributes. In addition to the many fruity aromas, mild floral notes were also perceived in the spirits. Particularly in the spirit made from the YC coffee pulp, a rose aroma was described for its odour and taste. The aromatic alcohol 2-phenylethanol, also known as phenethyl alcohol, is probably responsible for this. 2-Phenylethanol is an aromatic compound that occurs in several fermented foods and presents a rose-like odour [60]. The NMR spectroscopy analysis has shown that all three coffee pulp spirits contain phenethyl alcohol in a concentration of 1 g/hL pure alcohol. The sensory analysis also showed that the term *sweet* was the only descriptor that was mentioned for the taste of all three spirit samples.

Unfortunately, the two spirits made from YI and BT contained some distinct off-flavours. Probably the chemical odour (*adhesive* and *solvent*) was caused by a poor distillation technique or improper separation of the head, heart and tail fractions.

The sensory evaluation was performed about 10 weeks after the coffee pulp spirits were produced in El Salvador. The YI spirit was the only sample that was described as *pungent/alcoholic*. A pungent taste is a typical characteristic of fresh distillates. Maturation can be carried out to improve the organoleptic quality of spirits and reduce the pungent taste and odour [61].

In the acceptance test, the panelists rated the spirits from 0–5 in terms of the total performance. The most preferred spirit was made from the YC pulp and achieved an average of 3.13 ± 0.77 points. The fruit spirit with BT was rated with 2.96 ± 1.08 points and finally the YI was scored at 2.67 ± 1.21 points. Considering that the scale goes from 0–5, the three novel spirits are located in the decent middle range. Although no sample performed very well, the results can be summarised as satisfactory, despite the rather rudimentary distillation technology applied. As there were no off-flavours in the YC sample, this distillate was probably rated the most popular. The multiple-samples test revealed similar results as the acceptance test. The YI spirit scored the worst in both the attributes of odour and taste. Nevertheless, the other two spirits have been generally well rated; YC received the highest popularity for the odour, and BT for the taste. When looking at the mean values with the corresponding standard deviations, it becomes clear that each sample has its proponents but also its detractors.

There are articles published about the production of spirits from coffee by-products. Some authors also conducted sensory evaluations, but only a few describe the flavour profile. A distillate produced from spent coffee grounds obtained an acceptable organoleptic character with a smooth coffee flavour [62]. In a sensory analysis, a spirit from coffee pulp (*C. arabica* variety Catuaí 99 Vermelho) and wastewater achieved a higher acceptance for taste and aroma than a common sugarcane spirit. Additionally, and perhaps surprisingly, testers of the sensory evaluation described the aroma of the coffee pulp spirit as reminding them of brewed coffee [20]. Considering that the three spirits were produced in El Salvador, no coffee flavour was detected in any of the samples. This also corresponds to the expectations, because the typical coffee aroma is only created by roasting the green beans [63]. Yadav et al. [64] report that an alcoholic beverage made from coffee pulp got a superior taste than one made from mucilage.

There is a general lack of publications about the analysis of the organoleptic and chemical characteristics of coffee pulp spirits from different varieties. For this reason, it is difficult to compare our data, such as the flavour profile, with other findings. However, our three coffee pulp spirits demonstrate a pleasant, sweet and fruity flavour, especially with regard to the intense stone fruit.

## 5. Conclusions

Because the ethanol yield from the pulp mash was low and distillates from coffee fruit are almost unknown worldwide, it would be conceivable to market them as premium fruit spirits. Surely, there will be connoisseurs and gourmets who would pay a reasonable price for this specialty product.

The approach of transforming coffee cherries and coffee by-products into spirits for human consumption is not entirely new, it has just been neglected for a long time. An impressive indication of this is a passage from the French book CULTURE DU CAFÉIER by E. Raoul [59], which was published in 1897.

Currently, there is only one commercial spirit made from coffee pulp on the market today. The brand `Good Vodka` has a distillery in New York, USA. According to the information on their homepage, a concentrate from coffee pulp and wastewater from wet coffee processing is produced in Colombia. Afterwards, the mixture is shipped to the USA for the fermentation and distillation process. The manufacturers describe their coffee fruit spirit as sweet with hints of black pepper and vanilla [65,66]. Furthermore, there are certain manufacturers who offer distillates flavored with dried coffee husks, such as gin, vermouth or cascara liqueur. However, in these cases, the coffee by-products are not fermented and distilled. The cascara is used for maceration or a tea-like extract is added to the neutral alcohol. In summary, there are already a few alcoholic beverages with coffee by-products on the market, but there could certainly be more in the future.

It is desirable to stimulate research on coffee residues and thus develop new trends and processes. Not only the reduction of environmental damage, but also the socio-economic benefits for coffee farmers and their families are essential effects of the sustainable utilisation of coffee pulp. There is strong evidence that the use of coffee by-products has the potential to stabilise the economic security in coffee-growing regions. In the coffee value chain, the production of quality spirits from coffee by-products rich in fermentable sugars can generate significant economic benefits [20].

## Figures and Tables

**Figure 1 foods-11-01672-f001:**
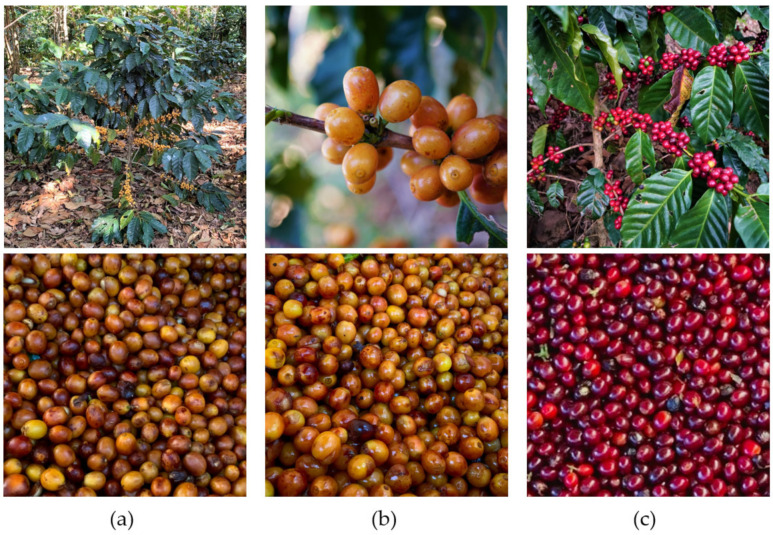
Photographs of the ripe coffee cherries before (top row) and after (bottom row) harvest of the varieties Yellow Colombia (**a**), Yellow Icatú (**b**) and Red Bourbon Tekisic (**c**).

**Figure 2 foods-11-01672-f002:**
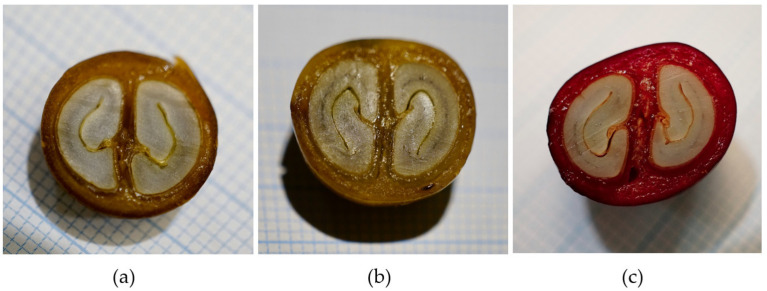
Photographs of cross-sections of coffee cherries (**a**) Yellow Colombia, (**b**) Yellow Icatú, (**c**) Bourbon Tekisic. Blue squares indicate a length of 1 mm.

**Figure 3 foods-11-01672-f003:**
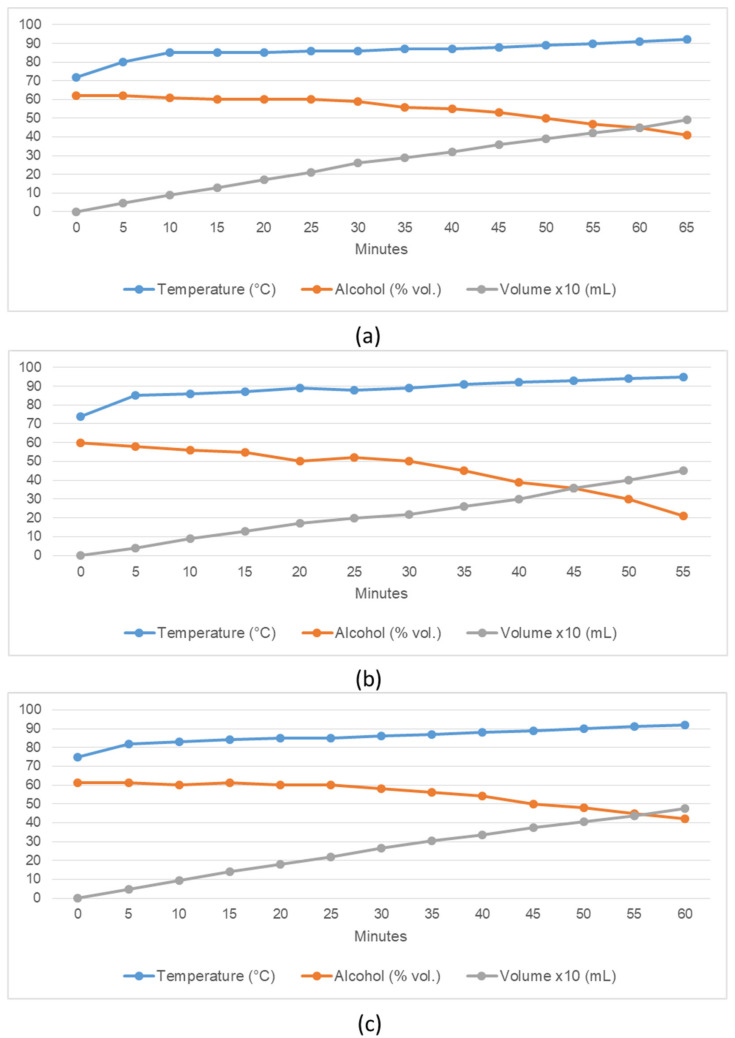
Temperature, alcoholic strength and volume curve of the second distillation run; (**a**) = YC, (**b**) = YI, (**c**) = BT.

**Figure 4 foods-11-01672-f004:**
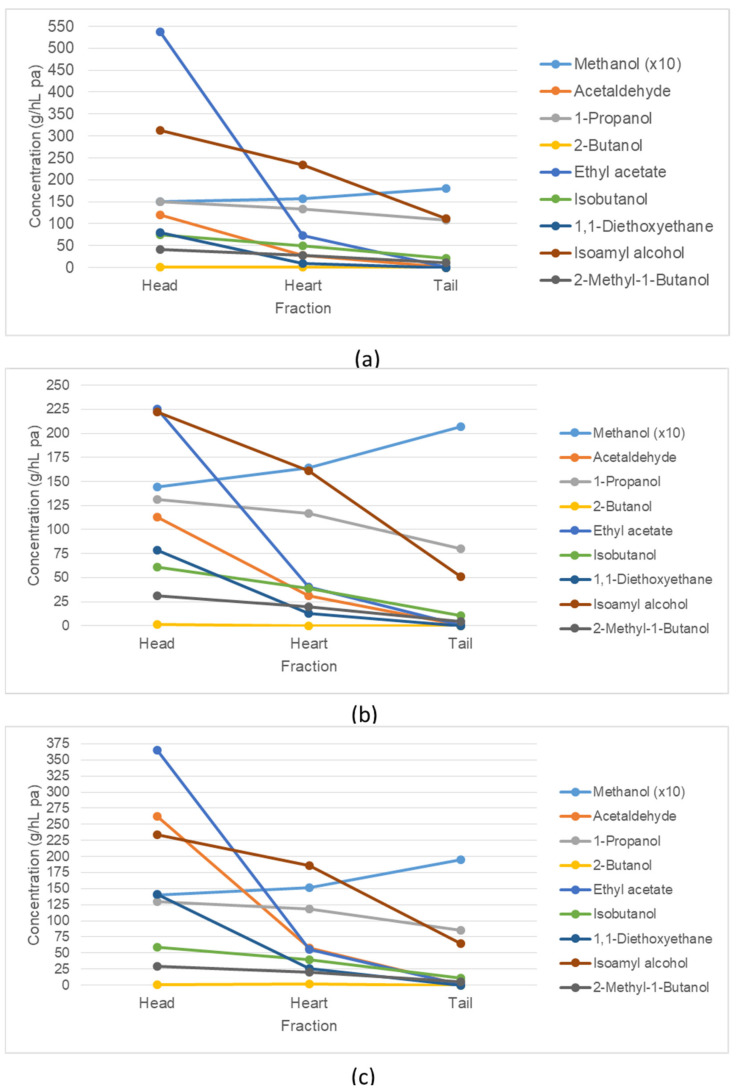
Volatile compounds in the different fractions of distillates, concentration in g/hL pa; (**a**) = YC, (**b**) = YI, (**c**) = BT.

**Table 1 foods-11-01672-t001:** Composition of the coffee pulp mashes.

	Yellow Colombia (YC)	Yellow Icatú (YI)	Red Bourbon Tekisic (BT)
Coffee pulp (kg)	17.0	10.0	17.0
Water (kg)	15.6	8.8	15.6
Yeast (g)	5.0	2.5	5.0
Enzyme (mL)	2.0	1.0	2.0

**Table 2 foods-11-01672-t002:** Substrate characteristics of coffee cherries (*n* = 10).

	Yellow Colombia (YC)	Yellow Icatú (YI)	Red Bourbon Tekisic (BT)
Weight (g)	1.31 ± 0.41	1.20 ± 0.28	1.68 ± 0.25
Length (cm)	1.35 ± 0.12	1.38 ± 0.11	1.53 ± 0.09
Diameter (cm)	1.22 ± 0.11	1.24 ± 0.10	1.36 ± 0.08
Pulp-to-fruit ratio (%)	51.6	56.8	54.6
Seed-to-fruit ratio (%)	47.5	41.7	44.3
Brix value (°Bx)	21.5 ± 0.5	24.0 ± 1.0	20.7 ± 1.5

**Table 3 foods-11-01672-t003:** Raw spirit results after the first distillation step (*n* = 1).

Mash	Yellow Colombia (YC)	Yellow Icatú (YI)	Red Bourbon Tekisic (BT)
Weight (kg)	32.6	18.8	31.1
Distillation Time (min)	142	110	184
Raw Spirit yield (mL)	3600	2700	3000
Alcoholic strength (% vol)	20	18	23

**Table 4 foods-11-01672-t004:** Yield of heads, hearts and tails after the second distillation run and the alcoholic strength of these fractions (*n* = 1).

		mL	g	% vol
Yellow Colombia (YC)	Head	150	125	70.1
	Heart	600	482	64.8
	Tail	238	178	51.9
Yellow Icatú (YI)	Head	70	51	70.6
	Heart	500	405	61.8
	Tail	292	145	37.4
Red Bourbon Tekisic (BT)	Head	92	66	73.1
	Heart	700	558	66.1
	Tail	278	215	43.7

**Table 5 foods-11-01672-t005:** Volatile compounds in the heart fractions of the distillates measured by NMR spectroscopy in g/hL pa.

	Yellow Colombia (YC)	Yellow Icatú (YI)	Red Bourbon Tekisic (BT)
Methanol	1412.7 ± 158.8	1456.7 ± 159.8	1403.7 ± 144.7
Ethyl acetate	61.7 ± 4.9	33.3 ± 6.5	49.7 ± 5.0
Acetaldehyde	15.3 ± 0.6	19.0 ± 1.0	33.7 ± 2.1
Isobutanol	66.7 ± 4.5	43.0 ± 14.0	48.0 ± 2.6
1-Propanol	128.7 ± 12.9	109.7 ± 14.2	118.7 ± 12.2
Isoamyl alcohol	210.3 ± 35.9	141.7 ± 32.1	180.0 ± 24.6
Phenethyl alcohol	1.0 ± 0.0	1.0 ± 0.0	1.0 ± 0.0
Ethyl lactate	2.0 ± 0.0	2.0 ± 0.0	2.0 ± 0.0

## Data Availability

All relevant data are within the paper and its appendices.

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
