# Peer review of "Production of Coffee Cherry Spirits from Coffea arabica Varieties"

_foods, 2022, doi:10.3390/foods11121672_

Round 1
Reviewer 1 Report
This manuscript describes the use of a residue from the coffee industry to be used in the production of a liquor with aromatic characteristics. This work is of great importance to increase the income of coffee producers and reduce the pollution that these residues generate in the producing regions.
The work is well written and its scientific contribution is solid, however, it is suggested to include some data that will allow a better understanding of the study:
Page 2, line 86 and 87: Indicate what criteria were taken into account for the selection of the yeast strain used. Mainly criteria such as resistance to ethanol, Killer effect, etc.
Page 4 line 132-136: Indicate the characteristics and conditions of the equipment used for the maceration of the coffee pulp.
Page 5 line 211: Specify the chromatographic separation conditions (type of column, volume, injection mode and temperature, flow rate and type of gas and detection temperature used in the study of the volatile fraction of distillates).
Page 8 line 308: there is an empty parenthesis.
Page 10 line 349: state or suggest a mechanism by which an increase in methanol was observed during fermentation
Author Response
This manuscript describes the use of a residue from the coffee industry to be used in the production of a liquor with aromatic characteristics. This work is of great importance to increase the income of coffee producers and reduce the pollution that these residues generate in the producing regions.
RESPONSE: Thank you for your assessment of our paper.
The work is well written and its scientific contribution is solid, however, it is suggested to include some data that will allow a better understanding of the study:
Page 2, line 86 and 87: Indicate what criteria were taken into account for the selection of the yeast strain used. Mainly criteria such as resistance to ethanol, Killer effect, etc.
RESPONSE: In this context, the major criterion was the low capacity to product pectin methylesterase. We have added this information into lines 86-87.
Page 4 line 132-136: Indicate the characteristics and conditions of the equipment used for the maceration of the coffee pulp.
RESPONSE: We have indicated the supplier of the fermentation equipment in line 125, and deleted some redundancies on the equipment. All equipment not specified was already available at the farm, with suppliers mostly unknown.
Page 5 line 211: Specify the chromatographic separation conditions (type of column, volume, injection mode and temperature, flow rate and type of gas and detection temperature used in the study of the volatile fraction of distillates).
RESPONSE: As this is not an analytical chemistry paper, and the method was previously published, we decided to make the text more concise rather than expand it, also considering the length of the paper. We also took the chance to make the second method, NMR, more concise to have coherence in this section.
Page 8 line 308: there is an empty parenthesis.
RESPONSE: Thank you for spotting this mistake. The parenthesis was deleted.
Page 10 line 349: state or suggest a mechanism by which an increase in methanol was observed during fermentation
RESPONSE: Line 349 is in the results section (3.3.), which should not include discussion or speculation. The mechanism of methanol formation is already later discussed in the corresponding section of the discussion (4.3): “The polysaccharide pectin is a main component of the cell walls of plants. The enzyme pectin methylesterase transforms pectin into pectic acid and methanol.” The kinetics of the process are influence by various parameters, including methylation degree, yeast type, temperatures, so that an increase during fermentation is the typically expected behavior.
Reviewer 2 Report
After carefully reading the article entitled "Production of coffee cherry spirits from Coffea arabica varieties", I would recommend a minor revision to this article. Some points need to be addressed:
Line 129: How much water that you add into the fruit flesh?
Line 137: How much sulphuric acid that you add for acidification?
Table 2 : Please mention the replication.
Figure 3 and 4: Please show the standard deviation in the graph. What is the Y axis title? Concentration?
Line 363: The Title of the Table is missing. Please show the standard deviation in the table. Please state also in the table caption, how many replications was the analysis.
Figure 5: What is the Y axis title? Value?
Table 4: Please show the standard deviation in the table. Please state also in the table caption, how many replications was the analysis.
As this research using sensory analysis by using human, please make sure that the researchers already have the ethical clearance document.
Author Response
After carefully reading the article entitled "Production of coffee cherry spirits from Coffea arabica varieties", I would recommend a minor revision to this article.
RESPONSE: Thank you for the careful review of our article.
Some points need to be addressed:
Line 129: How much water that you add into the fruit flesh?
RESPONSE: The amounts of coffee pulp, water, yeast and enzyme are already specified for all trials in Table 1 on page 4.
Line 137: How much sulphuric acid that you add for acidification?
RESPONSE: We did not add a specific amount, but used the amount necessary to lower the pH to 3, as controlled by a pH meter.
Table 2 : Please mention the replication.
RESPONSE: The number of replicates were added as requested.
Figure 3 and 4: Please show the standard deviation in the graph. What is the Y axis title? Concentration?
RESPONSE: The standard deviations, which are very low, would not lead to an improved degree of information. The y-axis title is clearly stated on all graphs in Figures 3 and 4: g/L.
Line 363: The Title of the Table is missing. Please show the standard deviation in the table. Please state also in the table caption, how many replications was the analysis.
RESPONSE: In our file, a title of table 3 is stated. Due to the time constrictions in El Salvador and the shortness of coffee picking of each variety, only one distillation experiment was conducted. Hence, we cannot show standard deviations. We have added n=1 to the table caption as requested.
Figure 5: What is the Y axis title? Value?
RESPONSE: As the different lines in Figure 5 have different y-axis units, the units are stated in the legend below the figure, i.e., °C for temperature, % vol for alcoholic strength, and mL for volume (to be multiplied with 10 to have the correct value, because of different scales).
Table 4: Please show the standard deviation in the table. Please state also in the table caption, how many replications was the analysis.
RESPONSE: See above, as only single experiments were possible, n=1. This was added to the table legend (but it was also obvious from the methods section).
As this research using sensory analysis by using human, please make sure that the researchers already have the ethical clearance document.
RESPONSE: According to the legislation in the EU and Germany, the affiliated institutes do not require ethical clearance for sensory analysis of foods. Nevertheless, we considered the IFST guidelines for ethical and professional practices for the sensory analysis of foods [1]. According to that, potential adverse effects for the assessors were excluded as only small amounts of spirit were taken into the mouth. The methanol amount in this portion was considerably below toxic levels, even assuming that the assessor would ingest the portion and not spit it out according to typical practices of testing spirits. All assessors had training in sensory analysis of foods, and regularly performed sensory testing of spirits in line of their normal duties. Furthermore, the CVUA Karlsruhe is externally accredited by the national accreditation body of the Federal Republic of Germany (Deutsche Akkreditierungsstelle, DAkkS) for sensoric analysis of foods. The CVUA Karlsruhe is also permanently permitted by German federal state law to conduct sensory testing of alcoholic beverages in its capacity as governmental control laboratory (see [3] and details in Lachenmeier & Monakhova [4]).
References:
[1] https://www.ifst.org/membership/networks-and-communities/special-interest-groups/sensory-science-group/ifst-guidelines
[2] https://www.dakks.de/de/akkreditierte-stelle.html?id=D-PL-18866-02-00
[3] Ministerium Ländlicher Raum: Verwaltungsvorschrift des Ministeriums Ländlicher Raum über die Dienstaufgaben und Zuständigkeitsbereiche der Chemischen und Veterinäruntersuchungsämter und des Staatlichen Tierärztlichen Untersuchungsamtes Aulendorf – Diagnostikzentrum [Administrative regulation of the Ministry of Rural Affairs regarding the official duties and jurisdiction of the Chemical and Veterinary Investigation Laboratories and the State Veterinary Laboratory Aulendorf - center of diagnostic investigations]. GABl 2000, 2000:358-359.
[4] Lachenmeier, D.W., Monakhova, Y.B. Short-term salivary acetaldehyde increase due to direct exposure to alcoholic beverages as an additional cancer risk factor beyond ethanol metabolism. J Exp Clin Cancer Res 30, 3 (2011). https://doi.org/10.1186/1756-9966-30-3